# Contribution of the L-Type Amino Acid Transporter Family in the Diagnosis and Treatment of Prostate Cancer

**DOI:** 10.3390/ijms24076178

**Published:** 2023-03-24

**Authors:** Xue Zhao, Shinichi Sakamoto, Jiaxing Wei, Sangjon Pae, Shinpei Saito, Tomokazu Sazuka, Yusuke Imamura, Naohiko Anzai, Tomohiko Ichikawa

**Affiliations:** 1Department of Urology, Chiba University Graduate School of Medicine, Chiba 260-8670, Japan; 2Department of Pharmacology, Chiba University Graduate School of Medicine, Chiba 260-8670, Japan

**Keywords:** prostate cancer, LAT1, LAT3, diagnosis, treatment

## Abstract

The L-type amino acid transporter (LAT) family contains four members, LAT1~4, which are important amino acid transporters. They mainly transport specific amino acids through cell membranes, provide nutrients to cells, and are involved in a variety of metabolic pathways. They regulate the mTOR signaling pathway which has been found to be strongly linked to cancer in recent years. However, in the field of prostate cancer (PCa), the LAT family is still in the nascent stage of research, and the importance of LATs in the diagnosis and treatment of prostate cancer is still unknown. Therefore, this article aims to report the role of LATs in prostate cancer and their clinical significance and application. LATs promote the progression of prostate cancer by increasing amino acid uptake, activating the mammalian target of rapamycin (mTOR) pathway and downstream signals, mediating castration-resistance, promoting tumor angiogenesis, and enhancing chemotherapy resistance. The importance of LATs as diagnostic and therapeutic targets for prostate cancer was emphasized and the latest research results were introduced. In addition, we introduced selective LAT1 inhibitors, including JPH203 and OKY034, which showed excellent inhibitory effects on the proliferation of various tumor cells. This is the future direction of amino acid transporter targeting therapy drugs.

## 1. Introduction

Tumor growth requires continuous nutritional support, of which essential amino acids (EAAs) are an important source [1]. The amino acid uptake was found to be higher in tumor tissue than in normal tissue [2]. The L-type amino acid transporter (LAT) family is a group of transmembrane transporter proteins composed of four members, LAT1, 2, 3, 4. The first two proteins belong to the solute vector family 7 (SLC7) and the latter two belong to SLC43. They are important pathways for essential amino acids to enter cells and are closely related to intracellular pathways [3,4,5,6].

The LAT family was upregulated in many tumors [7]. Prostate cancer (PCa) is the most frequent malignant tumor in men globally. In the United States, approximately 2.6 million new cases of PCa were identified each year, with an estimated 34,500 fatalities [8]. We primarily reviewed the relationship between the LAT family and PCa and analyzed the role of LATs in the diagnosis and treatment of PCa.

## 2. The Structure and Function of LATs

### 2.1. LAT1

LAT1 (SLC7A5) was discovered in 1998 [3,9]. It combines with 4F2hc (SLC3A2) heavy chain protein to form a transmembrane complex to play its biological role [10]. LAT1 is mainly responsible for the transport of large neutral amino acids. Generally, LAT1 is expressed in the normal human body mainly in the gastrointestinal mucosa, testicular support cells, ovarian follicular cells, pancreatic islet cells, and endothelial cells that act as inter-tissue barriers (blood-brain, blood-retina, and blood-follicular barriers) [11]. Many neurotransmitters and neuroactive compounds cannot cross the blood-brain barrier, and most of them are synthesized in the brain. Therefore, LAT1, as the precursor of neurotransmitters and neuroactive compounds, plays an important role in amino acid uptake through the blood-brain barrier [12]. As previously stated, LAT1 expression in normal organisms is limited to specific cells and tissues. Studies have also shown that LAT1 exists in cells with high proliferation and differentiation ability, such as embryos and T-lymphocytes. During embryonic development, the LAT1 of syncytial trophoblast cells plays an indispensable role in the development of the placenta, contributing to the exchange of amino acids between mother and fetus. Systemic LAT1 knockout results in defects in the placenta, which can be fatal to the embryo in the second trimester [13]. There is no clear research result on the role of LAT1 in T cell differentiation, but previous studies have shown that LAT1 is related to the metabolic process of T lymphocyte differentiation. LAT1 expression is low in intrinsic T cells, whereas T cells at the differentiation stage specifically increase LAT1 expression by T cell receptors (TCR), ensuring sufficient nutrients to react with the antigen [10,14].

LAT1 provides branched-chain amino acid (BCAA), especially leucine, to the mammalian target of rapamycin complex1 (mTORC1), a major control factor of cell proliferation. mTORC1 senses amino acid signals and promotes cell proliferation through multiple downstream effectors related to gene expression and metabolism [15]. Upstream of mTORC1 is a complex of GTPase-activating protein (GAP), activity toward Rags 1 (GATOR1) and GATOR2. GATOR1 functions as a mTORC1 antagonist and GATOR2 functions as a mTORC1 agonist. GAP inhibits mTORC1 in the absence of amino acids. Leucine transports into the cell by LAT1 binds to the leucine sensor sestrin2. The interaction between leucine and sestrin2 can activate GATOR2 and inhibit GATOR1, thus promoting the function of mTORC1 to achieve the purpose of cell proliferation [16,17]. LAT1 expression is upregulated in many cancers, such as breast [18], lung [19], colorectal [20], renal [21], bladder [22], prostate [23], and gliomas [24].

One of the purposes of large quantities of amino acid transport in LAT1 in cancer cells is to use BCAAs as biosynthetic materials for the metabolic reprogramming of cancer cells. Branched-chain amino acid transferase (BCAT) deaminates free BCAAs to generate the appropriate branched-chain keto acid (BCKA). BCAT2 transforms BCAA into BCKA within the mitochondria, which is subsequently catalyzed by metabolic intermediates and acetyl-CoA into the TCA cycle, which is used for energy production and fatty acid metabolism [25] (Figure 1).

Another function of LAT1 that promotes tumor formation is to inhibit cell-damaging T-cell control in the tumor microenvironment. In the serine pathway, tryptophan is converted into kynurenine (Kyn) by 2, 3-Deoxygenase (TDO) and indoleamine 2, 3-Dioxygenase (IDO). In the physiological state, TDO and IDO levels are negligible, but in the cancer state, both enzymes rise dramatically. Through the continuous function of TDO and IDO, large amounts of Kyn can be synthesized. Kyn is transported from cancer cells to T cells by LAT1. In T cells, Kyn binds to the aryl hydrocarbon receptor (AHR), which inhibits the anti-tumor immune response of T cells and promotes the proliferation of cancer cells [26].

### 2.2. LAT2

LAT2 (SLC7A8) was discovered in 1999 [4,27,28]. The structure of LAT2 is similar to that of LAT1, and it forms a transmembrane complex with 4F2hc for material transport [29]. However, LAT2 has a wider range of substrate specificity than LAT1, including polar uncharged amino acids and small neutral amino acids [4]. LAT2 commonly expresses in normal humans [11] and is associated with the development of amino-aciduria [30].

Both LAT1 and LAT2 consist of 12 transmembrane domains that form pathways for their substrates [29]. They bind to the heavy glycoprotein subunit 4F2hc via disulfide bonds. Although 4F2hc does not appear to have a direct substrate transfer function [31], it enables more stable localization of LAT1 and LAT2 on the plasma membrane [32]. Previous studies have suggested that 4F2hc lacks amino acid transport activity. Instead, the combined LAT1 and LAT2 units are the only ones with transport capacity [31]. But now, there are different views on it. 4F2hc acts as a molecular chaperone to enable LAT1 and LAT2 to become its final site on the membrane [32]. 4F2hc is required for the transport of LAT1 and LAT2 to the plasma membrane [9], where LAT1 and LAT2 are thought to determine the transport properties of heterodimers. At the same time, increased 4F2hc expression levels in many forms of cancer are associated with poorer prognosis in several studies [33,34,35,36] (Figure 2).

### 2.3. LAT3

LAT3 (SLC43A1) was first named POV1, “Prostate cancer Overexpressed gene 1”, and it was upregulated in prostate cancer as a gene of unknown function [37]. It was really discovered definitively in 2003 [5]. LAT3 is usually expressed in the liver, skeletal muscle, and pancreas [38]. Overall LAT3 expression is low in the kidney, but LAT3 expression is stronger in the apical plasma membrane of the podocyte foot processes. It shows that LAT3 is important for the development and maintenance of podocyte function and structure [39]. Another study has also shown that LAT3 expression is required for erythrocyte development to produce hemoglobin [40]. LAT3 can be upregulated in response to androgen, which is closely related to leucine uptake and cell proliferation in human prostate cancer cell lines [41].

### 2.4. LAT4

LAT4 (SLC43A2) was discovered in 2005 and identified by homology with LAT3 [6]. LAT4 is usually expressed in epithelial cells of the small intestine, proximal renal tubules, and thick ascending limbs [6]. However, in mouse models, LAT4 is expressed in the intestine, kidney, brain, white adipose tissue, testis, and heart, but not detected in the liver, showing differences from human expression [6]. The physiological function of LAT4 in these organs is still not fully understood. In LAT4 knockout mice, newborn mice are smaller than wild-type mice, suggesting that LAT4 is important for growth and development [42].

Unlike LAT1 and LAT2, the biological functions of LAT3 and LAT4 do not require binding to heavy chains and can exist independently. LAT3 and LAT4 are both sodium-dependent neutral amino acid transporters. The exact mechanism of transport for LAT3 and LAT4 is not well understood, but it is thought to involve a symport mechanism, in which the transport of amino acids is coupled to the uphill movement of sodium ions against their concentration gradient. The movement of ions and amino acids is driven by the energy generated from the electrochemical gradient established by the sodium/potassium ATPase. The difference between LAT3 and LAT4 lies in their substrate specificity and tissue distribution. LAT3 has a higher affinity for large neutral amino acids, such as leucine and isoleucine [5], while LAT4 has a higher affinity for small neutral amino acids, such as alanine and serine [6] (Figure 1 and Figure 2).

## 3. LATs and PCa

LATs promote PCa progression in several ways:Amino acid uptake:LATs increase the uptake of essential amino acids into cancer cells, which supports their growth and survival. After being delivered into cells, these amino acids are used to make proteins, nucleic acids, lipids, and ATP. Compared with normal cells, cancer cells have higher upregulation transporters (LATs), which can promote the entry of foreign amino acids into cells, and the stable acquisition of amino acids by cancer cells is important for cancer growth. By increasing the availability of amino acids, LATs can promote PCa cell proliferation and invasion.Activation of signaling pathways:LATs have been shown to activate various signaling pathways, including the mTOR pathway, which is involved in the regulation of cell growth, proliferation, metabolism, and survival. LAT1 [23] and LAT3 [41] are highly expressed in prostate cancer, providing branched-chain amino acids (BCAA) to the mammalian target protein of rapamycin complex (mTORC1), which senses amino acid signaling and promotes cell proliferation through multiple downstream effectors related to gene expression and metabolism [15]. Leucine, which enters the cell via LAT1, binds to the leucine sensor sestrin2. The interaction between leucine and sestrin2 can activate GATOR2 and inhibit GATOR1, thus promoting the function of mTORC1 and achieving the purpose of cell proliferation [16,17]. LATs can regulate mTOR activity by influencing the availability of essential amino acids, such as leucine, that activate the pathway.Drug resistance:In PCa cells resistant to antiandrogen therapy (ADT), the expression of some LATs is up-regulated, which may promote the progression of PCa to castration-resistant prostate cancer (CRPC) through androgen receptor variants [43]. Changes in the microenvironment induced by hormone deprivation therapy can alter the expression of LAT1 and LAT3. Reduced androgen receptor signaling may lead to decreased LAT3 expression and, as another consequence, increased LAT1 expression. Changes in the microenvironment induced by hormone deprivation therapy can alter the expression of LAT1 and LAT3. Decreased androgen receptor signaling may lead to decreased LAT3 expression and, as another consequence, the production of the AR-V7 variant, resulting in increased 4F2hc expression, and decreased leucine, resulting in increased LAT1 expression. The two form a dimer that eventually causes leucine to be re-transported into the cell to promote cancer cell proliferation. PCa is transformed into CRPC, which is resistant to ADT treatment.Promotion of angiogenesis:LATs have been implicated in the regulation of blood vessel formation (angiogenesis), which is essential for the growth and spread of PCa cells. Tumors grow and evolve through constant crosstalk with the surrounding microenvironment. New evidence suggests that angiogenesis and immunosuppression often occur together in response to this crosstalk [44]. For example, the expression of LAT1 was significantly correlated with the expression of VEGF, CD34, and microvascular density at the primary and metastatic sites [45,46,47,48]. VEGF and CD34 are factors related to angiogenesis. LAT1 can also mediate the angiogenesis of miR-126 on primary human pulmonary microvascular endothelial cells by regulating mTOR signaling [49].Others: Increase the uptake of amino acids by inducing hemoglobin maturation [40]. LAT3 expression is required for the development of red blood cells to produce hemoglobin. LAT3 can be upregulated under the action of androgens [41], which leads to the increase of hemoglobin development and increases the way for tumor cells to obtain nutrients from another side, thus promoting their proliferation.

Current studies on prostate cancer and LATs usually focus on LAT1 and LAT3. There are few studies on LAT2 and LAT4. Studies show the main leucine transporters are different in different stages of prostate cancer, especially in the castration-resistant stage of androgen receptor (AR) expression [23,50]. Studies have found that LNCaP cells mainly express LAT3, while LAT1 is mainly expressed in DU145 and PC-3 cells [23,50]. According to Rii’s study, LAT3 is highly expressed in LNCaP and C4-2 cells expressing AR, but hardly expressed in AR-negative PC3 and DU145 cells [51]. Changes in the microenvironment, such as starvation or hormone deprivation, can promote cancer formation and alter LAT1 and LAT3 expression. Reduced androgen receptor signaling may lead to decreased LAT3 expression and, as another consequence, increased LAT1 expression [41].

In addition to the common relationships mentioned above, we will describe the relationship between LAT1, 2, 3, 4, and prostate cancer respectively.

### 3.1. LAT1

It has been found that the up-regulation of LAT1 during antiandrogen therapy (ADT) promotes the progression of PCa cells. LAT1 is highly expressed in CRPC cell lines. LAT1 knockdown significantly reduces cell proliferation, migration, and invasion. High LAT1 expression is associated with poor biochemical recurrence-free periods in patients treated with chronic ADT [23]. Another study also confirmed that LAT1 expression is up-regulated at the protein and mRNA levels in 22Rv1 CRPC tumors with chronic ADT [52]. Sugiura demonstrated a potential relationship between AR-V7 and the LAT1-4F2hc complex. AR-V7 activates downstream target genes in the absence of androgens. 4F2hc is one of the downstream target genes of AR-V7. The expression level of 4F2hc in CRPC tissues is significantly increased, which correspondingly suggests poor prognosis of related patients [43]. Furthermore, a study shows ATF4 gene expression is upregulated during metastasis, suggesting that ATF4-mediated amino acid response element-containing gene regulation may be important for the development of metastatic CRPC [53]. ATF4-regulated genes, such as LAT1 and 4F2hc, show low expression in normal prostate tissue and primary prostate cancer, but they are significantly increased in metastatic prostate cancer, suggesting that these transporters are involved in the nutrient supply required for metastatic prostate cancer [53] (Figure 3).

### 3.2. LAT2

LAT2 is reported to be less distributed in malignant tumors except for neuroendocrine tumors [55] and more distributed in normal tissues [3]. Therefore, LAT2 may be used as a reference marker for normal benign tissue or as an indication of a good prognostic outcome. One study [56] examined LAT1-4 expression and its association with clinical outcomes in a combined cohort of more than 18,000 radical prostatectomy specimens. The expression of LAT1-3 in prostate cancer is higher than that in benign tissues except for LAT4. The expressions of LAT2, LAT3, and ASCT2 are negatively correlated with GS ≥ 8, lymph node invasion, and high Decipher score. The lowest decile of LAT3 and ASCT2 expression correlates with the worst MFS. LAT2 and LAT3 expression is associated with better clinical outcomes [56]. y + LAT2 (SLC7A6) is an alternative light subunit that constitutes the cationic and neutral amino acid heterodimer transport system y + L. A study [50] established the castration-resistant prostate cancer (CRPC) model LN-cr with androgen AR expression. Compared with LNCaP, y + LAT2 expression is increased in LN-cr. These results suggest that androgen removal induces the down-regulation of LAT3 and up-regulation of y + LAT2 in LNCaP cells.

### 3.3. LAT3

It is highly expressed in primary PCa [5]. Previous studies have shown that LAT3 expression is reduced in metastatic and/or castration-resistant cancers; therefore, LAT3 expression may be associated with androgen dependence in PCa [53]. In another word, LAT3 is highly expressed in prostate cancer cells that expressed the androgen receptor (AR). LAT3 is highly expressed in LNCaP and C4-2 cells that expressed AR but hardly expressed in PC3 and DU145 cells without AR. LAT3 mediates leucine uptake in LNCaP and PC3 cells [41]. Its expression is increased under the treatment of dihydrotestosterone and reduced under bicalutamide treatment [51]. Growth factors such as EGF activate the PI3K/Akt/mTORC1 signaling pathway, which regulates different protein synthesis programs leading to cell growth. This pathway relies on mTORC1 to detect adequate levels of intracellular amino acids, which inversely regulates LAT3 expression to enhance amino acid access [57]. LAT3 knockdown inhibits phosphorylation of mTOR, eukaryotic translation initiation factor 4EBP1, and ribosomal protein S6K1, but does not inhibit phosphorylation of Akt. And AR knockdown results are similar (Figure 3). Furthermore, as above mentioned, erythrogenesis involves increased uptake of neutral essential amino acids through LAT3 [40]. As red blood cells mature, their transcription profile changes, reflecting altered metabolic states, including induction of genes for iron and heme metabolism, as well as those involved in the amino acid cycle. The mRNA expression of LAT1 and LAT3 is significantly increased in mature red blood cells, but no 4F2hc expression is detected, suggesting that LAT1 may have no transport function [40]. The high expression of LAT3 in prostate cancer will undoubtedly promote the uptake of more amino acids by red blood cells on the other hand.

### 3.4. LAT4

There is currently little research on LAT4 and prostate cancer. A study [52] has shown that LAT4 expression is up-regulated in CRPC cell lines. Another study has found that 18F-labeled amino acids, such as 3-O-methyl-6-18F-fluoro-L-dopa (18F-OMFD) and 18F-fluorodihydroxyphenylalanine (18F-FDOPA), are important imaging agents for PET in vivo tumor display [58,59]. 18F-OMFD appears to be a suitable diagnostic imaging tracer for amino acid transport in poorly differentiated squamous cell head and neck carcinoma with increased LAT1 and LAT4 expression [59]. Similarly, 18F-OMFD and 18F-FDOPA should also have diagnostic values in CRPC with high LAT4 expression [52]. A study found that the expression of LAT4 increases after amino acid ingestion in mouse models treated with N-butyl- (4-hydroxybutyl) nitrosamine (BBN) [60]. However, further studies and evidence on the value of LAT4 in the diagnosis and treatment of prostate cancer are lacking.

Table 1 summarizes the specific relationship between LATs and prostate cancer, as well as the corresponding inhibitors (Table 1).

## 4. Prostate Cancer Diagnosis by LATs

### 4.1. LAT1

Among amino acid transporters, LAT1 is selectively hyperactive in a variety of cancer cells [10]. The pathway for enhancing LAT1-mRNA expression is not yet clear but includes carcinogenic Myc and hyposia-inducedfactors (HIFs) that can enhance its expression. That suggests the feedforward mechanism of tumor formation [64,65]. LAT1 can be used as a tumor marker for prostate cancer, and LAT1 expression is highly correlated with high proliferation index, stage, and poor prognosis [23,66].

In addition to being a tumor marker, LAT1 will play an important role in the diagnosis as a transporter. In other words, LAT1 selectively delivered drugs can be used for the diagnosis of PCa. LAT1 special use matrix of cancer diagnosis of positron emission computed tomography (PET) is a powerful technology for clinical prostate cancer detection. PET works by detecting the radioactive isotopes labeled on the tracer to find out where the tracer accumulates. Cancer cells accumulate PET tracers, which mimic the nutrients they need to proliferate. (18) F-labeled fluoroalkyl phenylalanine derivatives, as PET tracers, are more likely to bind to LAT1 in tumors, which demonstrates the effectiveness of 18f-labeled aromatic side chain pet tracer in the diagnosis of prostate malignancies [67]. The U.S. Food and Drug Administration approved trans-1-amino-3-18f-flucyclobutane carboxylic acid (anti-[18F]-FACBC) PET to detect prostate cancer in patients with elevated prostate-specific antigen after treatment in 2016 [68]. The utility of LAT1 in PET imaging has been demonstrated in clinical practice.

At present, Japan is further developing a new PET tracer based on LAT1 delivery, the NKO series. The goal is to adopt a simpler and more efficient 18f markup structure. At present, the clinical study in the normal human body has been completed [69].

### 4.2. LAT3

As mentioned above, the expression level of LAT3 is different in different stages of prostate cancer. Studies have shown that LAT3 is regulated by androgen receptors. LAT3 is significantly reduced in CRPC, and LAT3 is also reduced after androgen deprivation therapy. LAT3 is expected to be a tumor marker to judge the progression of prostate cancer from HSPC to CRPC [41,43,51,53].

### 4.3. LAT4

As mentioned above, 18F-OMFD has diagnostic value as a PET tracer in CRPCS with high LAT4 expression [59].

## 5. Inhibitors of LATs and Targeted Therapy of PCa

Inhibitors that target transporters include compounds that transport as a substrate and non-transport compounds that act as blockers or exert an inhibitory effect on isosteroids. As transport inhibitors, the current mainstream view of pharmacology believes that non-transport compounds are superior to transport compounds because non-transport compounds do not accumulate in cells and have high affinity [69]. 2-aminobicyclo-(2,2,2,1)-heptane-2-caryboxylic acid (BCH) (Figure 4(Aa)) is a nonmetabolic leucine analogue. BCH is a specific inhibitor of the sodium-independent L system (LAT1, LAT2, LAT3, and LAT4). In the study of LATs, BCH is widely used [53,70]. However, because BCH is delivered as a substrate and has a low specificity and affinity for LAT1, it has not been used at the clinical level. Clinical studies on anti-cancer drugs (inhibitors) targeting LATs mainly focus on LAT1 and LAT3.

In LAT1, the earliest reported specific LAT1 inhibitor was the thyroid hormone triiodothyronine (T3) (Figure 4(Bb)), which showed a high inhibitory effect (low Ki value) and was almost non-transportable molecule [71], thus providing an idea for the research and development of LAT1 inhibitors. Therefore, JPH203 (KYT-0353) (Figure 4(Cc)) was successfully developed in 2009 as a non-delivery LAT1-specific inhibitor (LAT1 blocker) [72]. The core structures of T3 and JPH203 both contain an amino acid backbone and a bulky side chain (Figure 4(Bb,Cc). Then, JPH203 has recently been widely studied as a potent inhibitor of LAT1. JPH203, like T3, has a high affinity and is specific to LAT1, but does not affect LAT2. JPH203 interferes with the constitutive activation of mTORC1 and Akt, reduces c-MyC expression, and triggers a folding protein response mediated by CHOP transcription factors associated with cell death [73]. Since then, several studies have confirmed that JPH203 has a significant inhibitory effect on the growth of common tumor cells. A Phase I clinical study found that JPH203 is well tolerated and provided positive prognostic outcomes in the treatment of biliary tract cancer, with a disease control rate of approximately 60% against biliary tract cancer [74]. The authors are currently planning Phase I and II studies of JPH203 in CRPC. Although JPH203 has been shown in multiple studies to inhibit leucine uptake by tumor cells and has shown concentration-dependent cytotoxicity in vitro or performed well in transplanted tumor models, the human phase I clinical trial is a milestone. In addition, a Japanese research team has developed an SKN series of LAT1 inhibitors similar to T3 and JPH203 (Figure 5(Aa)) [75,76]. In recent years, a research team led by Professor Kanai at Osaka University is developing the OKY series of novel LAT1 inhibitors. Among OKY compounds, OKY-034 shows a high inhibitory effect and specificity against LAT1. The above amino acid LAT1 inhibitors are competitive inhibitors, but OKY-034 has a non-competitive inhibitor style because the structure of OKY-034 does not include the amino acid skeleton. The advantage of non-competitive inhibitors is that only small amounts (low concentrations) are required to show effect, due to the need to react competitively with the endogenous amino acid matrix. In addition, OKY-034 does not require large hydrophobic sites such as T3 and SKN, so it is relatively soluble and can be taken orally. Phase I/IIa trial of OKY-034 safety and efficacy in patients with pancreatic cancer is being conducted at Osaka University Hospital (UMIN000036395) [69]. It is believed that these drugs will soon be used to treat prostate cancer.

However, compared with LAT1, there is less evidence to support the general role of LAT2 in cancer, and it is more likely to be used as a benign biomarker. Since there is no known drug substrate or inhibitor targeting LAT2 except for BCH, unfortunately, no studies have been conducted on LAT2 as a tumor-targeted therapy. However, LAT2’s innate ability to transport amino acids makes it a potential target for the diagnosis or treatment of prostate cancer based on amino acids.

In terms of LAT3, LAT3 knockdown inhibits cell proliferation, migration, invasion, and phosphorylation of p70S6K and 4EBP-1 [51]. Studies have shown that ESK242 and ESK246 are effective inhibitors of LAT3 (Figure 5(Bb)). ESK246 preferentially inhibits leucine transport through LAT3, while ESK242 can inhibit both LAT1 and LAT3. Its use in prostate cancer cells further suggests that ESK246 is a potent leucine uptake inhibitor that leads to decreased mTORC1 signaling, cyclin expression, and cell proliferation [63]. New anti-prostate cancer therapies targeting LAT3 may build on this.

## 6. Conclusions

The L-type amino acid transport family has valuable clinical significance in malignant tumors represented by prostate cancer. LAT1 is not only a reliable tumor biomarker but also an imaging tracer target. LAT1 and LAT3 can be used in the diagnosis and prognosis of prostate cancer. Moreover, the specific expression levels of LAT1 and LAT3 have the value of judging the development of prostate cancer to castration-resistant prostate cancer, which can effectively guide the anti-androgen therapy of prostate cancer and predict the possibility of biochemical recurrence. In addition, LAT1 and LAT3 are also significant therapeutic targets for prostate cancer. Several inhibitors targeting the corresponding LATs are in clinical trials and are expected to be widely used in the targeted therapy of prostate cancer in the near future.

## Figures and Tables

**Figure 1 ijms-24-06178-f001:**
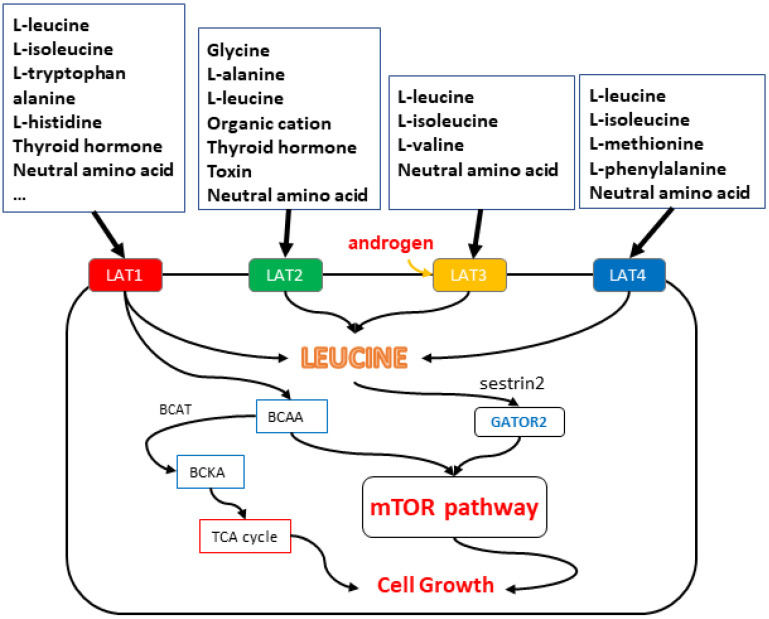
The substances that each LAT is mainly responsible for transporting are shown in the figure. LATs provide branched-chain amino acids (BCAA), especially leucine, to mammalian target cells of rapamycin complex 1 (mTORC1), and the mTOR pathway is a major control factor in cell proliferation. Leucine is transported into the cell by LATs, binds to the leucine sensor sestrin2, mTORC1 senses amino acid signal, and GATOR2 acts as mTORC1 agonist. Thus, it can promote the function of the mTOR pathway and achieve the purpose of cell proliferation. In addition, LAT1 can use BCAAs as biosynthetic materials for the metabolic reprogramming of cancer cells. BCAT deaminates free BCAAs to form BCKA. It enters the TCA cycle and is used for energy production and fatty acid metabolism to promote cell proliferation.

**Figure 2 ijms-24-06178-f002:**
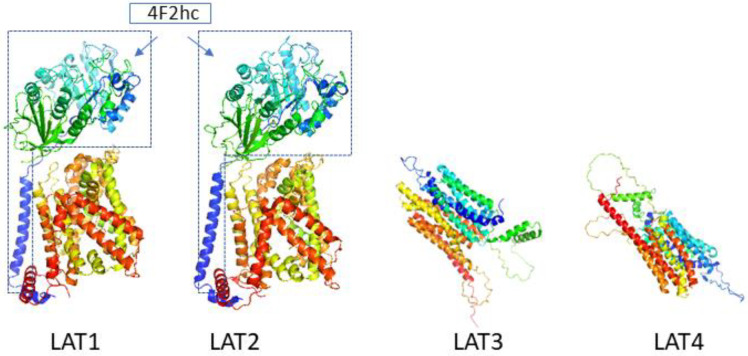
The three-dimensional conformation of LATs. Both LAT1 and LAT2 consist of 12 transmembrane domains that form pathways for their substrates. They bind to the heavy glycoprotein subunit 4F2hc via disulfide bonds. Unlike LAT1 and LAT2, the biological functions of LAT3 and LAT4 do not require binding to heavy chains and can exist independently. (Images created using PyMol*, colord by chainbows, the PDB ID: 6IRS, 7CMI, the Uniprot ID: O75387, Q8N370, PyMol* (version 2.5 Schrodinger. Warren L. DeLano), and RCSB PDB, and UniProt).

**Figure 3 ijms-24-06178-f003:**
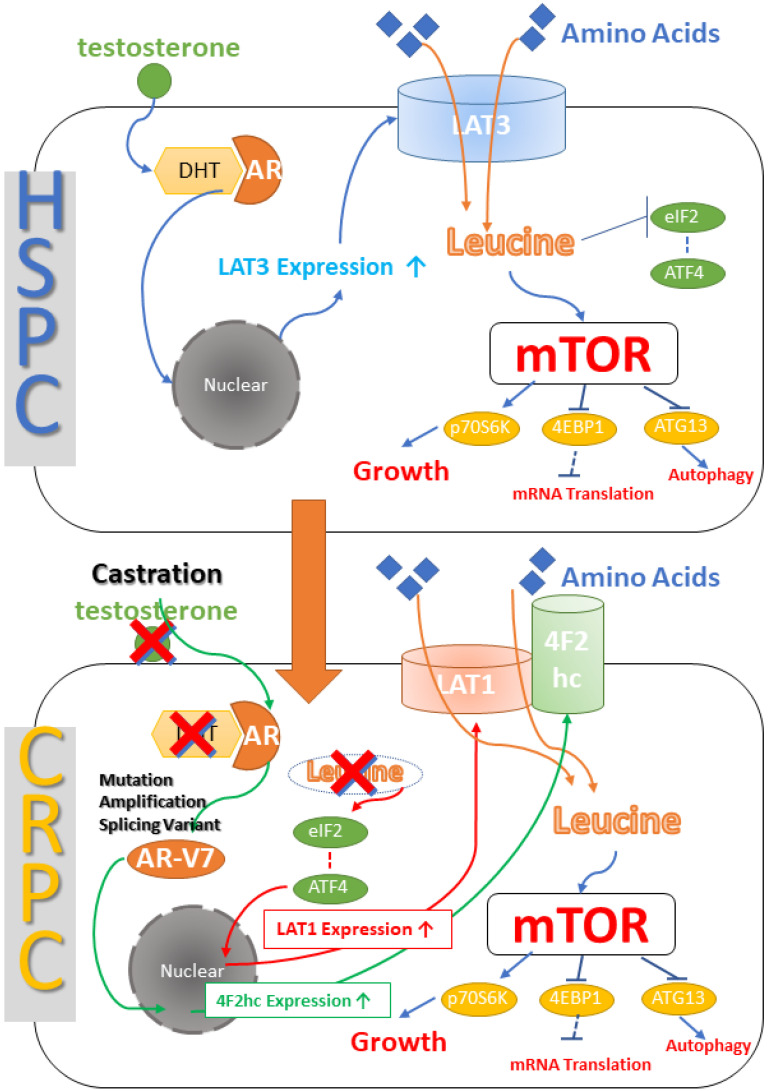
LAT is linked to both hormone-sensitive prostate cancer (HSPC) and castration-resistant prostate cancer (CRPC). In untreated HSPC, 5-alpha reductase converts testosterone to dihydrotestosterone (DHT), which binds to the androgen receptor (AR), enters the nucleus, and stimulates LAT3 transcription, resulting in enhanced LAT3 expression and contributes to the mTOR pathway activation. When hormone therapy is used to treat PCa, testosterone levels fall, resulting in castration, and ARs that no longer bind DHT change, increase, and generate splicing variants. AR-V7, in particular, can enter the nucleus in the absence of testosterone activation, and 4F2hc is present in its downstream signaling. Furthermore, the removal of leucine from the cells results in the lack of eIF2 repression and the admission of ATF4 into the nucleus. It enhances LAT1 expression, and LAT1 and 4F2hc form a dimer, allowing leucine into the cell and promoting tumor cell proliferation [54].

**Figure 4 ijms-24-06178-f004:**
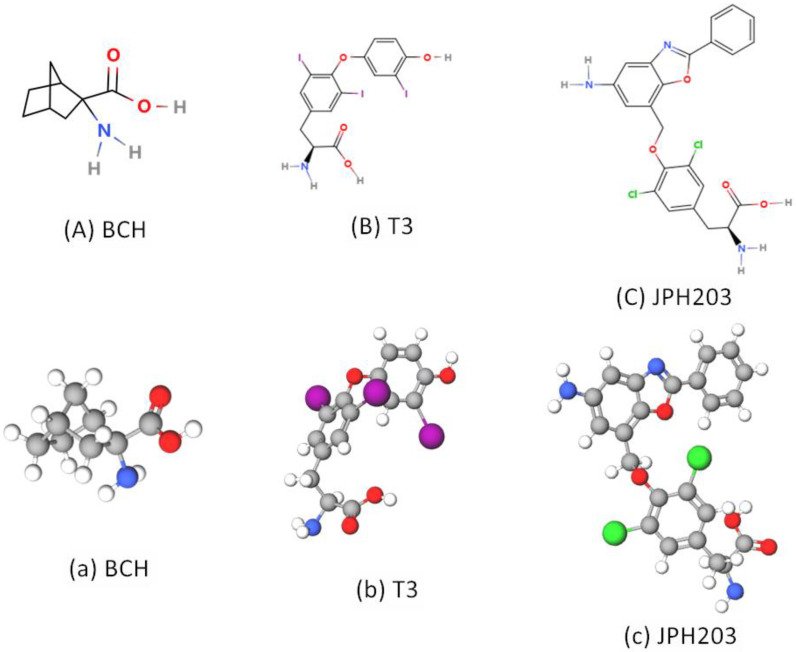
The 2D (**A**–**C**) and 3D (**a**–**c**) structures of LATs inhibitors. (**Aa**) BCH is a transportable system L inhibitor. (**Bb**,**Cc**) The core structures of T3 and JPH203 both contain an amino acid backbone and a bulky side chain.

**Figure 5 ijms-24-06178-f005:**
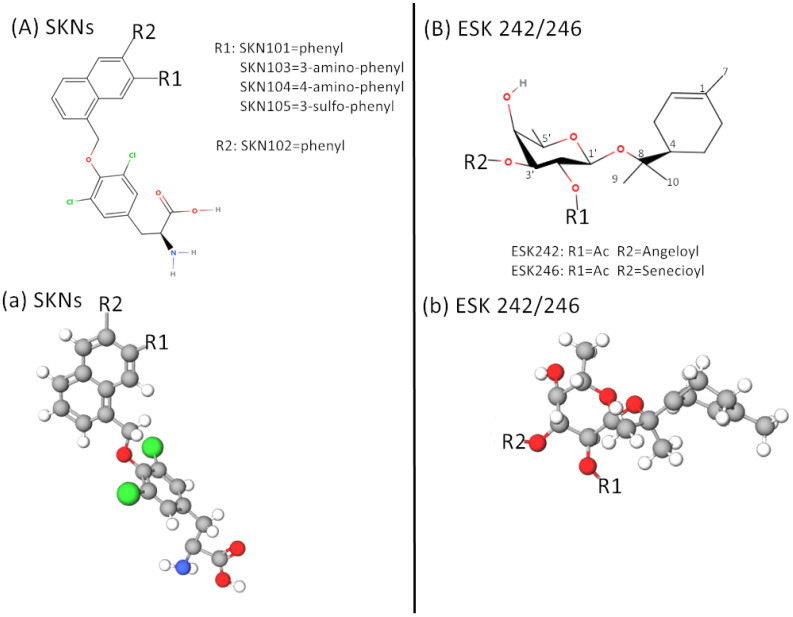
The 2D (**A**,**B**) and 3D (**a**,**b**) structures of LAT1 inhibitor SKN series and LAT3 inhibitor ESK series. (**Aa**) SKN series and JPH203 have similar molecular structures. (**B**,**b**) The ESK series shows a different molecular structure from LAT1 inhibitors such as T3, JPH203, and SKNs. ESK242 can inhibit both LAT1 and LAT3.

**Table 1 ijms-24-06178-t001:** The current relationship between LATs and prostate cancer. (Symbol ‘—’ represents that this cell line does not express the corresponding LATs according to the citation paper).

LATs	PCa Cell Lines	Up-Regulation of Expression	Inhibitors	Be Inhibited Effects	Diagnosis/Treatment
LAT1 [23,33,41,43,50,53,61]	LNCAP	↑/—	T3, BCH, JPH 203, ESK242, SKN, OKY-034	Lower proliferation, Higher apoptosis, Lower leucine absorption, Lower mTORC1 activity, Amino acid stress, Reduced tumor metastasis ability.	Used as a PET tracer transporter in the diagnosis of malignant tumors. As a target for targeted therapy.
	C4-2	↑			
	PC3	↑			
	DU145	↑			
	VCAP	↑			
	22Rv1	↑			
LAT2 [55,56]	prostate specimen	↑	BCH	N/A	Associated with a better prognosis.
LAT3 [5,41,51,53,54,62,63]	LNCAP	↑	ESK242, ESK246	Lower proliferation, higher apoptosis, Reduced tumor metastasis ability.	As a tumor marker for HSPC to CRPC transformation, As a targeted therapeutic target.
	C4-2	↑			
	PC3	—			
	DU145	—			
LAT4 [42,52,58,59]	22Rv1 (Simulate the CRPC situation)	↑	N/A	Growth retardation in mouse models.	Possible as a PET tracer target for 18F-labeled amino acids in CRPC.

## Data Availability

Not applicable.

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
