# Peer review of "Contribution of the L-Type Amino Acid Transporter Family in the Diagnosis and Treatment of Prostate Cancer"

_ijms, 2023, doi:10.3390/ijms24076178_

Round 1

Reviewer 1 Report

The review focuses on the role of L-type amino acid transporter (LAT) family members - LAT 1-4 in prostate cancer.  The LATs are involved in various metabolic pathways and have been reported to promote the progression of prostate cancer by increasing amino acid uptake, activating the mTOR pathway and downstream signals, mediating castration resistance, promoting tumor angiogenesis, and enhancing chemotherapy resistance. The research in the domain is relatively in nascent stages and the review sheds light on the critical role of LATs and their clinical significance and diagnostic/ therapeutic application. 

1. Although the review is well-articulated, a need to correct tenses and grammar in the manuscript is felt. The authors may do the needful in this respect.

2. Section-3 (LATs and PCa), where the ways in which LATs promote PCa progression are briefed in points may be elaborated rather than summarizing the study in a single line. This would make the review more comprehensive and interesting to read.

Author Response

  1. Although the review is well-articulated, a need to correct tenses and grammar in the manuscript is felt. The authors may do the needful in this respect.

A: We thank the Reviewer #1 for carefully reviewing and kindly providing very helpful comments to improve our manuscript.

We reviewed the paper again and corrected the mistakes in tense and grammar in the paper.

  1. Section-3 (LATs and PCa), where the ways in which LATs promote PCa progression are briefed in points may be elaborated rather than summarizing the study in a single line. This would make the review more comprehensive and interesting to read.

A: We thank the Reviewer #1 for carefully reviewing and kindly providing very helpful comments to improve our manuscript.

As suggested by reviewers, we extended this paragraph summarizing the main ways that LATs promote the proliferation and development of prostate cancer. The mechanism of LATs that may be involved in each approach is explained in detail (Line193-243). In addition to the common mechanism, more detailed relationships are described one by one in the following paragraphs in the paper.

Reviewer 2 Report

The review introduced basic knowledge of LATs and their roles in the development of cancer, which suggests their potential as therapeutic targets. 

I have some suggestions based on the contents:

1. Author needs to improve figure 1, what is the difference between the dotted line and solid lines? There are so many different fonts and sizes that make me confused. Also, the author should introduce more details about the figure in the figure legends. A similar revision should be made to figure 3.

2. In figure 2, the authors tried to compare the structures of LATs, but they did not do a good job. Please revise the orientation and color codes of each structure and predicted model, making them uniform and more easy to compare. Also, the highlighted membrane lines and squares make me even more confused and did not get the point that author try to express.

To make a better comparison,  I suggest the author align all structures together in pymol, then show each of them in the same color codes and same angle so that the helixes will be easier to compare with each other. Even though they may be in different conformations, part of the domain should be usable for align.   

3. For section 5, I suggest the author add a figure of the chemical structures of LATs inhibitors. 

Author Response

  1. Author needs to improve figure 1, what is the difference between the dotted line and solid lines? There are so many different fonts and sizes that make me confused. Also, the author should introduce more details about the figure in the figure legends. A similar revision should be made to figure 3.

A: We thank the Reviewer #2 for carefully reviewing and kindly providing very helpful comments to improve our manuscript.

We redrew Figure1, adjusted the font, rearranged the location of the paths, and removed some misleading ICONS, such as the dotted line (which originally indicated that LEU was part of the BCAA). We described the figure legends of Figure1 and Figure3 in more detail and added references (Line 123 & Line 258), hoping to make the pictures easier to understand.

  1. In figure 2, the authors tried to compare the structures of LATs, but they did not do a good job. Please revise the orientation and color codes of each structure and predicted model, making them uniform and more easy to compare. Also, the highlighted membrane lines and squares make me even more confused and did not get the point that author try to express.

To make a better comparison,  I suggest the author align all structures together in pymol, then show each of them in the same color codes and same angle so that the helixes will be easier to compare with each other. Even though they may be in different conformations, part of the domain should be usable for align.   

A: We thank the Reviewer #2 for carefully reviewing and kindly providing very helpful comments to improve our manuscript.

We redrew Figure2, we removed the misleading diagram of the cell membrane. Because the transmembrane structures of LAT3 and LAT4 on the cell membrane were not very clear, we could not effectively compare them with those of LAT1 and LAT2. Therefore, we deleted the three views and remade the 3D structure drawing of LATs with Pymol, using chainbows for all LATs’ color modes. The detailed transmembrane structure of LAT3 and LAT4 is the direction of our further research.

      3.For section 5, I suggest the author add a figure of the chemical structures of LATs inhibitors. 

A: We thank the Reviewer #2 for carefully reviewing and kindly providing very helpful comments to improve our manuscript.

We added the molecular formula and 3D molecular structure diagram of the LATs inhibitors quoted in the paper as Figure 4 (Line 378) and Figure 5 (Line 415). And the pictures are annotated where the corresponding inhibitors are mentioned. We are sorry that we have not found the molecular formula of OKY034, possibly for reasons of patent and research confidentiality.
